# Lipid Peroxidation and Iron Metabolism: Two Corner Stones in the Homeostasis Control of Ferroptosis

**DOI:** 10.3390/ijms24010449

**Published:** 2022-12-27

**Authors:** Luc Rochette, Geoffrey Dogon, Eve Rigal, Marianne Zeller, Yves Cottin, Catherine Vergely

**Affiliations:** 1“Pathophysiology and Epidemiology of Cerebro-Cardiovascular Disease” Research Unit (PEC2, EA 7460), University of Burgundy and Franche-Comté, UFR des Sciences de Santé, 7 Boulevard Jeanne d’ Arc, 21000 Dijon, France; 2Cardiology Unit, CHU-Dijon, 21000 Dijon, France

**Keywords:** ferroptosis, ferritinophagy, iron, ferritin, lipid peroxidation, autophagy

## Abstract

Regulated cell death (RCD) has a significant impact on development, tissue homeostasis, and the occurrence of various diseases. Among different forms of RCD, ferroptosis is considered as a type of reactive oxygen species (ROS)-dependent regulated necrosis. ROS can react with polyunsaturated fatty acids (PUFAs) of the lipid (L) membrane via the formation of a lipid radical L• and induce lipid peroxidation to form L-ROS. Ferroptosis is triggered by an imbalance between lipid hydroperoxide (LOOH) detoxification and iron-dependent L-ROS accumulation. Intracellular iron accumulation and lipid peroxidation are two central biochemical events leading to ferroptosis. Organelles, including mitochondria and lysosomes are involved in the regulation of iron metabolism and redox imbalance in ferroptosis. In this review, we will provide an overview of lipid peroxidation, as well as key components involved in the ferroptotic cascade. The main mechanism that reduces ROS is the redox ability of glutathione (GSH). GSH, a tripeptide that includes glutamic acid, cysteine, and glycine, acts as an antioxidant and is the substrate of glutathione peroxidase 4 (GPX4), which is then converted into oxidized glutathione (GSSG). Increasing the expression of GSH can inhibit ferroptosis. We highlight the role of the x_c_^-^ GSH-GPX4 pathway as the main pathway to regulate ferroptosis. The system x_c_^-^, composed of subunit solute carrier family members (SLC7A11 and SLC3A2), mediates the exchange of cystine and glutamate across the plasma membrane to synthesize GSH. Accumulating evidence indicates that ferroptosis requires the autophagy machinery for its execution. Ferritinophagy is used to describe the removal of the major iron storage protein ferritin by the autophagy machinery. Nuclear receptor coactivator 4 (NCOA4) is a cytosolic autophagy receptor used to bind ferritin for subsequent degradation by ferritinophagy. During ferritinophagy, stored iron released becomes available for biosynthetic pathways. The dysfunctional ferroptotic response is implicated in a variety of pathological conditions. Ferroptosis inducers or inhibitors targeting redox- or iron metabolism-related proteins and signal transduction have been developed. The simultaneous detection of intracellular and extracellular markers may help diagnose and treat diseases related to ferroptotic damage.

## 1. Introduction

Programmed cell death (PCD) plays a fundamental role in animal development and tissue homeostasis. Developing tissues and organs rely on an elaborate balance between cell division and PCD to reach appropriate cell numbers. In many organs, such as the central nervous system (CNS), cells are overproduced and subsequently removed by PCD. It has been estimated that more than half of all neurons generated in the mammalian CNS are eliminated by PCD. During development, several structures that operate a transitory function are removed by PCD when they are no longer required [1,2]. Many recent studies have focused on increasing our understanding of the basis of the development and aging at the cellular and molecular levels in various organs. It is known that there is an age-related decline in cellular quality control pathways such as autophagy and mitophagy, leading to the accumulation of potentially harmful cellular components in cells. Regeneration is a process of regrowth or repair with the capacity to maintain homeostasis, even after severe injury [3]. Depending on the type of tissue damage, the regenerative process includes several steps, including wound repair. Identifying the molecular mechanisms of degenerative changes, cell death, regeneration, and rejuvenation is important for developing therapies to treat various diseases. Cell death was divided into three main categories, based on distinctive morphological features: type I cell death, referred to as apoptosis; type II cell death, referred to as cell death involving autophagy; and type III cell death, referred to as necrosis. Autophagy plays a multifaceted role in regulating both the quality and quantity of protein and organelles (e.g., mitochondrial number and function). The induction of autophagy has been generally considered a programmed cell survival mechanism in response to various types of stress.

Recently it has been demonstrated that there are multiple forms of regulated cell death, which are different in their molecular mechanisms and morphological characteristics. New biological findings led to the concept of ferroptosis. This novel form of regulated cell death (RCD), distinct from apoptosis and necrosis was coined in 2012 [4]. Ferroptosis is focused by iron-dependent lipid peroxidation, which takes place in phospholipids, and is carried out in a second step by a radical chain reaction. In this context, ferroptosis is dependent on the peroxidation of phospholipids with polyunsaturated fatty acyl parts. The induction of ferroptosis involves specific pathways: redox-active iron, iron-dependent peroxidation enzymes, and antioxidant pool: glutathione-glutathione peroxidase 4 (GPX4). Genetic studies performed in cells and mice established the selenoenzyme glutathione peroxidase (GPX4) as the key regulator of this form of cell death. Using inducible GPX4(−/−) mice, a major study elucidated the essential role for the glutathione/GPX4 axis in preventing lipid-oxidation-induced acute renal failure and associated death. A potent spiroquinoxalinamine derivative: liproxstatin-1 is able to suppress ferroptosis in cells, in Gpx4 −/− mice, and in a pre-clinical model of ischemia/reperfusion-induced hepatic damage [5]. Many ferroptosis modulators are directly or indirectly related to GPX4 function. Sensitivity to ferroptosis is linked to intracellular and extracellular redox conditions. Accumulating evidence indicates that ferroptosis requires the autophagy machinery for its execution [6].

Intracellular iron accumulation and lipid peroxidation are two central biochemical events leading to ferroptosis. Multiple organelles, including mitochondria and lysosomes, are involved in the regulation of iron metabolism and redox imbalance in ferroptosis [7,8,9,10]. In this review, we provide a critical analysis of (1) the current molecular regulatory networks of ferroptosis and (2) the potential physiological functions of ferroptosis, together with the potential for therapeutic targeting. The connection between ferroptosis and cancer progression has been validated using ferroptotic agents. Ferroptosis has attracted considerable attention because of its potential role as a target for novel therapeutic anticancer strategies.

## 2. Mechanisms Governing Ferroptosis

### 2.1. The Process of Lipid Peroxidation

Ferroptosis is characterized by the production of reactive oxygen species (ROS) and lipid peroxidation. Oxidative stress can be described as an imbalance between the production of oxidant species and the antioxidant defenses, which may affect cellular redox homeostasis leading to molecular alterations and thus resulting in cell damage. The term “oxidants” is a general term used to identify several groups of reactive molecules among which ROS and nitrogen reactive species (RNS). A free radical is defined as any atom or molecule possessing unpaired electrons. Molecular oxygen, O_2_, is a biradical with two unpaired electrons. ROS production mainly occurs in mitochondria although they may have other locations (such as NADPH oxidase). Mitochondrial ROS production depends in an interweaving way upon many factors such as the membrane potential, the cell type and the respiratory substrates, ROS are singlet electron intermediates formed during the partial reduction of oxygen to water and they include radical and non-radical intermediates. The biologically relevant free radicals derived from oxygen are the superoxide anions (O_2_ ^•−^), and the hydroxyl radical (•OH). ROS arise in the course of normal cellular life, especially during cellular respiration. These highly reactive species are controlled by a protective system both enzymatic and non-enzymatic, which helps to prevent the accumulation of peroxidative damage to the cell. RNS include nitrogen compounds such as nitric oxide (•NO), nitrogen dioxide (NO2•), nitrate (NO3^−^), nitrite (NO2^−^), and peroxynitrite (ONOO^−^). ROS/RNS are natural byproducts of aerobic metabolism and are produced by all living multicellular organisms. ROS/RNS are produced by healthy cells in a highly regulated fashion in order to maintain the intracellular redox homeostasis [11]. As will be developed later, iron is an essential trace element, distributed in various subcellular organelles and the level of intracellular or mitochondrial ferrous iron (Fe^2+^) is increased in ferroptotic situations. The levels of iron in cells are controlled by a complex process. In addition to iron-triggered ROS production by the Fenton reaction, mitochondria- or NADPH oxidase (NOX)-mediated ROS production also play a cell type-dependent role in initiating lipid peroxidation that involves the metabolism of iron [12]. Ferroptosis is a highly regulated ROS-dependent type of cell death, derived from free iron overload (FIO) [13].

Lipid peroxidation is a process under which oxidants such as free radicals or nonradical species attack lipids containing carbon–carbon double bond(s), especially polyunsaturated fatty acids (PUFAs) that involve hydrogen abstraction from a carbon, with oxygen insertion resulting in lipid peroxyl radicals and hydroperoxides. Lipids are essential components of cell membranes that maintain structure and control the function of cells. They are primary targets of the attack by ROS. Cellular membranes contain significant amounts of PUFAs that are esterified on phospholipids, as well as free cholesterol. PUFAs are a family of lipids with two or more double bounds, that can be classified in omega-3 (n-3) and omega-6 (n-6) FAs according to the location of the last double bond relative to the terminal methyl end of the molecule. The principal n-6 FA is arachidonic acid (AA). It is a precursor of enzymatic and non-enzymatic oxidized products. These lipids are the primary targets for free radical attack. Lipid peroxides result from the reaction of oxygen derived free radicals with polyunsaturated FAs of membranes phospholipids. There are a number of reviews of lipid peroxidation that have appeared in last years, particularly related to the biology of lipid oxidation products [14,15].

There are two ways in which lipid peroxidation is triggered: non-enzymatically or enzymatically. Non-enzymatic lipid peroxidation is driven by radicals. Glycolipids, phospholipids (PLs), and cholesterol are also well-known targets of oxidative damaging. Lipids also can be oxidized by enzymes such as lipoxygenases (LOXs) and cyclooxygenases (COXs). There are six isoforms of LOXs in human. LOX enzymes are non-heme iron-containing enzymes, and the Fe^2+^ in the catalytic center requires oxidation to Fe^3+^ for LOX activation. LOX activation is coupled to the cellular redox state including to glutathione (GSH) levels. An important component of the defense against ROS is the capacity of the cell to buffer the redox environment. The redox environment of higher eukaryotic cells is defined by the redox state of numerous redox couples including reduced glutathione/oxidized glutathione (2GSH/GSSG), reduced thioredoxin/oxidized thioredoxin protein-SH/protein-SS-R and ascorbate/dehydroascorbate. The role of these redox couples in cellular processes has been reviewed extensively by us and other groups over the last few decades [16,17,18].

It has been reported that lipid peroxidation products may interfere in vivo with several biological processes, such as substrate-receptor interaction, signal transduction, gene expression, and homeostatic responses. A variety of transcription factors may be activated as a result of oxidative stress, leading to the expression of various genes, including those for growth factors, inflammatory cytokines, chemokines, cell cycle regulatory molecules, and anti-inflammatory molecules. A new concept of immunogenic cell death has recently been proposed. The immunogenic characteristics of this cell death mode are mediated mainly by damage-associated molecular patterns (DAMPs), most of which are recognized by pattern recognition receptors. The ferroptosis of non-immune cells will lead to the release of DAMP and initiate immune cell responses. The molecular pathways involved in translocation of calreticulin to the cell surface and secretion of ATP from tumor cells. Ferroptosis affects different types of immune cells in the cases of tumor, inflammation, and infection providing courses for targeted therapeutic interventions [19,20].

### 2.2. Steps of Lipid Peroxidation

The mechanism of free radical autoxidation involved in the process of lipid peroxidation consists of three steps: initiation, propagation, and termination steps. In the initiation step, the start is the formation of a lipid radical L•. Cellular membrane lipid oxidation can be induced by (1) exogenous physical and chemical inductors such as UV light, or ionization radiations, and (2) various endogenous enzymatic systems generating ROS. Concerning the second step: propagation; molecular oxygen is added to the carbon-centered radical L• to generate a peroxyl radical (LOO•). The next propagating step in free radical chain oxidations is the transfer of a hydrogen atom from the organic substrate to a chain-carrying peroxyl radical (L-O-OH +L•). In the termination reaction, antioxidants provide a hydrogen atom to the LOO• species and form an antioxidant-radical that reacts with another LOO• forming non-radical products.

A recent report found that phospholipids containing two PUFA tails are particularly effective drivers of ferroptosis, suggesting that lipid crosslinking may be an aspect of the membrane damage upon ferroptosis [21]. PUFAs are the site of oxidative lipid damage and are required for the execution of ferroptosis. The abundance of PUFAs determines the extent of available lipid peroxidation sites and thus ferroptosis susceptibility [22]. ROS can react with PUFAs of the lipid membrane and induce lipid peroxidation to form L-ROS. High concentrations of L-ROS can trigger oxidative stress in cells, causing oxidative damage. Various diseases are associated with oxidative stress and lipid peroxidation, which leads to the accumulation of lipid peroxidation-derived reactive aldehydes and may consequently cause an increase in the formation of ROS/RNS and/or a decrease in the antioxidant defense. Several classes of antioxidant agents may be considered but it is important to clarify some points concerning the specificity of each antioxidant agent. An antioxidant can be defined as any substance that, when present in very low concentrations compared to that of an oxidizable substrate, significantly delays or inhibits the oxidation of that substrate. Defense mechanisms against free radical-induced oxidative stress involve: (1) preventive mechanisms, (2) repair mechanisms, and (3) antioxidant defenses. To prevent the interaction between radicals and biological targets, the antioxidant should be present at the location where the radicals are being [12].

### 2.3. Electrophilic Stress and Lipid-Derived Electrophiles

As we reported previously, AA is a precursor of enzymatic and non-enzymatic oxidized products. Among these compounds: prostaglandins, thromboxanes, leukotrienes, lipoxins, and isoprostanes have varied biological actions that may exert signaling or damaging roles during physiological and pathological conditions. Recent data support the concept that COXs, LOXs, and cytochrome P450 (CYP450) followed by cytosolic and microsomal dehydrogenases can convert AA to lipid-derived electrophiles (LDE) [23].

Concerning the molecular mechanisms, the molecule that supplies the electron pair is a nucleophile and the reactant that accepts the electron pair is an electrophile. In biology, there are two major sources of electrophiles: via external or endogenous ways. Compounds derived from oxidation of PUFAs such as 4-hydroxynon-2-enal (4-HNE) and malondialdehyde (MDA) contribute to electrophilic stress. 4-HNE is generated from lipid hydroperoxides of ω-6 fatty acids including linoleic acid and arachidonic acid. MDA is an end-product generated by decomposition of AA and larger PUFAs through enzymatic or non-enzymatic processes. Various methods have been applied to the measurement of 4-NHE and MDA. MDA has been measured by thiobarbituric acid (TBA) test and has been intensively studies as the lipid peroxidation end products in relation with cellular functions in pathological states. The role of these compounds in cellular processes has been reviewed extensively by us and other groups over the last few decades [12,24,25].

Other oxidation products generated from free radical lipid peroxidation, and some of the oxidation products such as the F2-isoprostanes, have stability in tissues and biological fluids. Plasma oxidative stress was determined by hydroperoxides (ROOH) and the ascorbyl radical/ascorbate ratio [26,27,28].

## 3. Iron Homeostasis

### 3.1. Iron Metabolism

As we reported in the first part of this review, ferroptosis is focused by iron-dependent lipid peroxidation. In this context, ferroptosis is dependent on the peroxidation of phospholipids with polyunsaturated fatty acyl parts in relationship with iron homeostasis. The induction of ferroptosis involves specific pathways: redox-active iron, iron-dependent peroxidation enzymes. Iron metabolism is also regulated primarily by the liver, which maintains systemic iron balance by producing and secreting factors, regulators of iron homeostasis. Several proteins, such as transferrin (Tf), ferritins, hepcidin, and ferroportin (FPN) exert crucial functions in the maintenance of systemic iron homeostasis [29]. The hepcidin–FPN axis is the principal regulator of extracellular iron homeostasis in health and disease [30].

Iron is a fundamental cofactor for several enzymes involved in oxidation–reduction reactions due to its ability to exist in two ionic forms: ferrous (Fe^+2^) and ferric (Fe^+3^) iron. However, the redox ability of iron can lead to the production of oxygen free radicals, which can damage various cellular components. For this reason, iron levels in tissues must be tightly regulated. Macrophages play an important role in executing the regulatory events that lead to changes in systemic iron levels. Cellular iron homeostasis is accompanied by the coordinated regulated expression of ferritin and other proteins including Tf. Cells have Tf receptors (TfR) that mediate iron metabolism. Iron is imported through the endocytosis of Fe^3+^-loaded Tf that interacts with its TfRs (TfR1 and TfR2) in a tightly regulated feedback control. Iron regulatory proteins1 and 2 (IRP1 and IRP2) record cytosolic iron concentrations and post-transcriptionally regulate the expression of iron metabolism genes [31,32].

Ferritins and hepcidin play a major role in iron homeostasis, cellular functions and control of cell death in relationship with redox control. Ferritin deficiency also induces ferroptosis via downregulation of solute carrier family 7 member 11 (SLC7A11), the inhibition of which triggers ferroptosis [33]. Studies show that p53 inhibits cystine uptake and sensitizes cells to ferroptosis by repressing expression of *SLC7A11*. The tumor suppressor p53 stated as the ‘guardian of the genome’ is mutated in approximately 50% of all cancers. It was the first to be linked to increased sensitivity to ferroptosis. Inactivation of the p53 tumor suppression pathway is a critical place in the formation of most human cancers. The tumor suppression activity of p53 is implicated in cell-cycle arrest, apoptosis and/or senescence in response to cellular stress. Stress-induced activation of p53 protein is mainly realized by post-translational modifications [34].

As will be discussed later in this review, the primary cellular defense pathway against ferroptosis is the SLC7A11 (p53)–GPX4 signaling axis, wherein GPX4 uses GSH as its cofactor to quench lipid hydroperoxides. SLC7A11 imports cystine for GSH biosynthesis. SLC7A11 overexpression facilitates GSH synthesis to suppress ferroptosis, suggesting an important role for a ferritin-SLC7A11-GSH axis in the therapy against ferroptosis-mediated diseases [33,34,35].

As we reported, several proteins, such as Tf, ferritins, and hepcidin exert crucial functions in the maintenance of systemic iron homeostasis. Hepcidin is an antimicrobial peptide with iron regulatory properties and it plays a significant role in iron regulation in health and disease. Hepcidin is predominantly synthesized in hepatocytes, secreted from hepatocytes and excreted through the kidney. Hepcidin is an 84-amino-acid (aa) preprohormone, and is targeted to the secretion pathway by a 24-aa N-terminal targeting sequence [30,31,32,33,34,35,36]. Various of studies reveal that hepcidin and Tf contribute to ferroptosis regulation and ferritinophagy process.

### 3.2. Ferritinophagy: A Selective Autophagic Degradation of Ferritin (Figure 1)

The term ferritinophagy is used to describe the removal of the major iron storage protein ferritin by the autophagy machinery. In this context, three iron storage proteins are known: ferritin heavy chain1 (FtH1), ferritin light chain (FtL), and mitochondrial ferritin (FtMT). Translation of FtH and FtL is under the control of the IRP process. Nuclear receptor coactivator 4 (NCOA4) is a cytosolic autophagy receptor used to bind ferritin for subsequent degradation by ferritinophagy. Ferritinophagy, a selective form of autophagy, contributes to the initiation of ferroptosis. During ferritinophagy, stored iron is released and becomes available for biosynthetic pathways.

**Figure 1 ijms-24-00449-f001:**
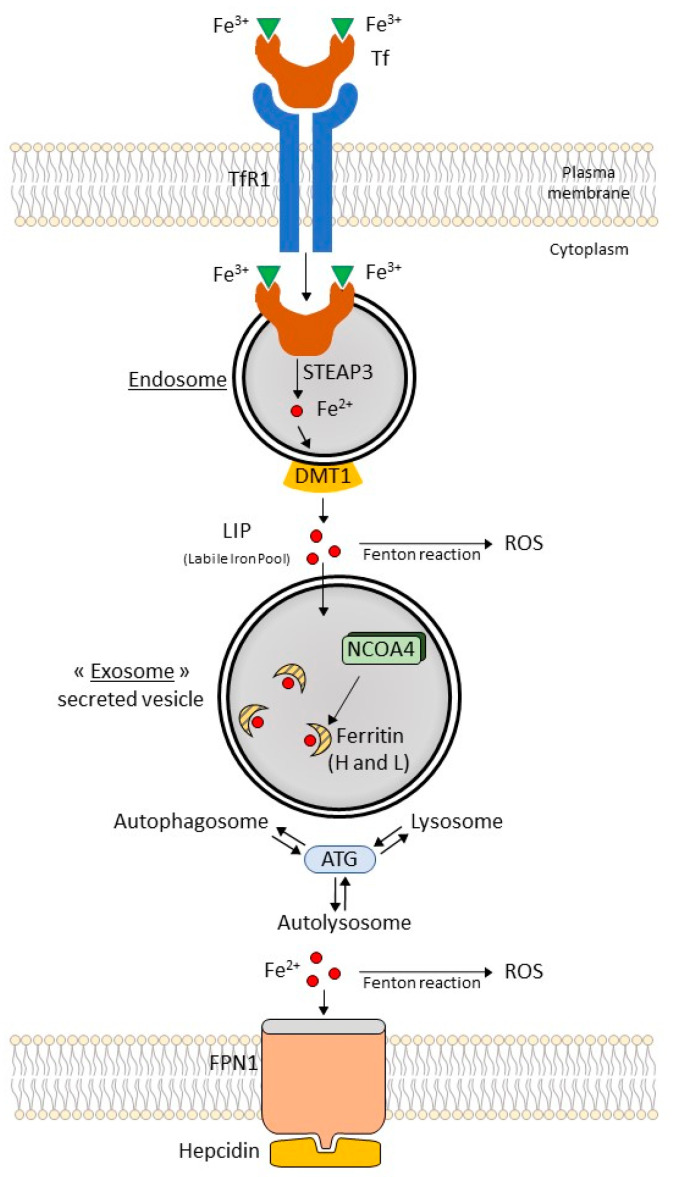
Ferritinophagy and iron metabolism. Extracellular Fe^3+^ binds to transferrin (Tf) and is taken up into cells through transferrin receptor 1 (TfR1). Fe^3+^ is reduced to Fe^2+^ by STEAP3 (six-transmembrane epithelial antigen of prostate 3); metalloreductases localized in the endosome. Divalent metal transporter 1 (DMT1) mediates the transport of Fe^2+^ from the endosome to a labile iron pool (LIP) in the cytoplasm. Fe^2+^ produces reactive oxygen species (ROS) via the Fenton reaction. In exosomes, excess iron from the LIP is stored in ferritin heteropolymers (ferritin heavy chain 1: H or ferritin light chain: L). Ferritin degradation is mediated by nuclear receptor coactivator 4 (NCOA4). The NCOA4 binds ferritin mediating its autophagic degradation in the process of ferritinophagy. Ferritinophagy is associated with interactions between autophagosomes, autolysosomes, and lysosomes, which are regulated by autophagy-related (ATG) proteins. Ferroportin 1 (FPN1) exports intracellular Fe^2+^; the hepcidin–FPN1 axis is a regulator of extracellular iron homeostasis.

Macrophages are known for their central role in iron metabolism. Under normal circumstances, macrophages play an important part in the recycling of iron through the engulfment of red blood cells (RBCs). As the number of RBCs suddenly rises or the RBCs are seriously damaged, increased erythrophagocytosis will happen. The regulation and functional consequences of ferritinophagy in relationship with macrophages properties remain to be defined [37,38].

Under iron-deprived conditions, NCOA4-mediated ferritinophagy contributes to the maintenance of mitochondrial functions through iron release within the cell and supply to mitochondria [39]. Ferritinophagy refers to selective autophagic degradation of ferritin, leading to the buildup of cytosolic iron in the form of ferrous iron (Fe^2+^), culminating in ferroptosis. As we reported, there is an age-related decline in cellular quality control pathways such as autophagy and mitophagy. Mitophagy is an important process in relationship with ROS metabolism. Mitophagy is the selective degradation of mitochondria through autophagy, which plays a role in maintaining the quantity and quality of mitochondria. NCOA4-mediated ferritinophagy has been shown to induce ferroptosis by degrading ferritin and inducing iron overload. A cytosolic iron chaperone poly(rC)-binding protein 1 (PCBP1) is a multifunctional RNA-binding protein involving gene transcription, RNA regulation, and iron loading to ferritins. PCBP1 known to repress autophagy is required for preventing iron-mediated toxicity and preventing ferroptosis. Taken together, these studies indicate that iron metabolism is tightly regulated at multiple levels through the ferroptosis pathway. Ferritinophagy and its regulation by NCOA4 has implications in health and disease. Sepsis evokes NCOA4-dependent ferritinophagy and increased cytosolic Fe^2+^ content [40,41,42,43].

## 4. Regulation of Ferroptosis

### 4.1. Cellular Pathways and Control of Ferroptosis (Figure 2)

In addition to the mechanisms that regulate the processes related to lipid peroxidation and iron homeostasis, ferroptosis is controlled by specific genes interventions. Among the mechanisms governing ferroptosis in relationship with important cellular pathways; we will describe the key role of cellular components, system x_c_^-^ and GPX4.

**Figure 2 ijms-24-00449-f002:**
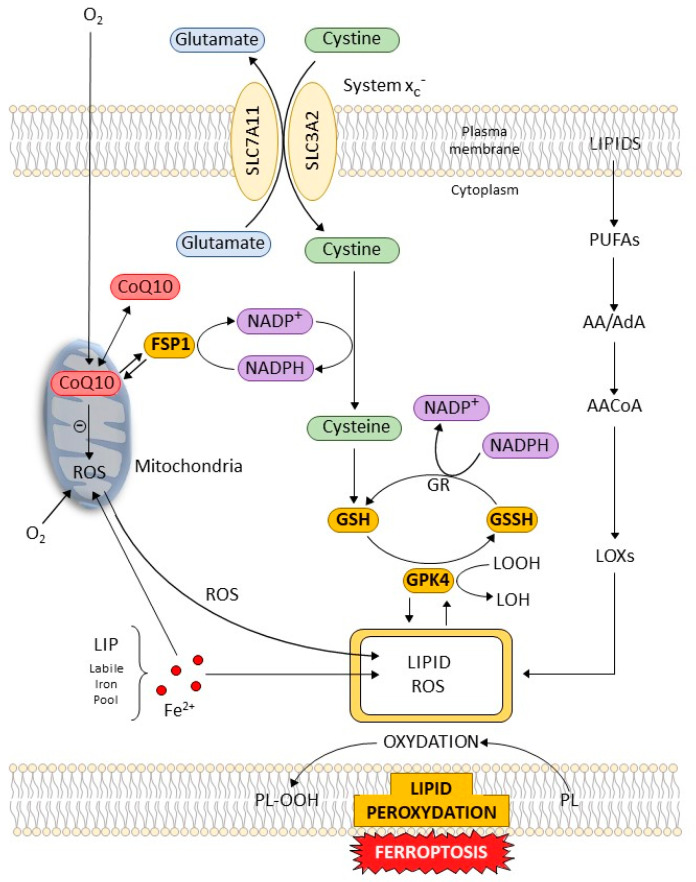
Regulatory pathways of ferroptosis. System x_c_^-^ functions as a cystine/glutamate antiporter that imports one molecule of cystine in exchange for one molecule of intracellular glutamate. System x_c_^−^ is a heterodimer containing the light chain subunit SLC7A11 and the heavy chain subunit SLC3A2. Cystine imported into the cell is converted to cysteine through a NADPH-NADP^+^ reaction. Reduced glutathione (GSH) is oxidized to oxidized glutathione (GSSG) GSSG is then converted back to GSH via glutathione reductase (GR). Glutathione peroxidase 4 (GPX4) uses GSH to reduce lipid hydroperoxides (LOOH) to lipid alcohols (LOH), thereby suppressing ferroptosis labile iron pool (LIP) produces LIPID-reactive oxygen species (ROS) via the Fenton reaction and through the lipoxygenase (LOX) pathway reaction and subsequent lipid peroxidation. Polyunsaturated fatty acid, especially arachidonic acid (AA) and adrenic acid (AdA), are the primary targets for free radical attack. Phospholipids (PL) in cell membranes and in lipoproteins can be oxidized to hydroperoxides (PL-OOH). The mitochondria is a major source of cellular ROS and Coenzyme Q10 (CoQ10) systems limit oxidative damage. The factor ferroptosis suppressor protein 1 (FSP1) is implicated in the FSP1-CoQ10-NADPH pathway. FSP1 reduces CoQ10 to ubiquinol, which, in turn, limits ROS production and lipid peroxidation. LIP accumulation, ROS production, and lipid peroxidation are the central biochemical events leading to ferroptosis.

The cyst(e)ine–GSH–GPX4 axis is considered the main system that opposes ferroptosis in mammals. The GSH/GPX4 axis plays an essential role in counteracting the production of specific phospholipid hydroperoxides in the presence of catalytically active iron. The upstream component in this system is system x_c_^-^: a cystine–glutamate antiporter composed of the transporter protein SLC7A11 linked via a disulfide bond to the regulatory subunit SLC3A2. Inhibition of system x_c_^-^ activates cellular ferroptosis in cell lines due to a lack of intracellular cysteine [44]. System x_c_^-^ is an amino acid transporter on the cell membrane that imports cystine and exports glutamate, leading to GSH synthesis [45]. Inhibiting SLC7A11, the subunit of system x_c_^-^, by drugs, e.g., erastin; the action leads to GSH depletion and the subsequent inactivation of GPX4, thereby causing lipid peroxidation-mediated ferroptotic cell death.

Central cellular signaling pathways and cellular contacts are determinants of ferroptosis. The plasma membrane allows the cell to sense and adapt to changes in the extracellular environment by relaying external inputs. Among these pathways, the nuclear factor E2-related factor 2/antioxidant response element (Nrf2-ARE) pathway is among the most important cell defense and survival pathways. This system is a key factor for cell protection from oxidative and electrophilic insults. The Keap1-Nrf2-ARE (Kelch-like ECH-Associating protein 1) pathway is closely associated with inflammatory disease. Among these transcription factors, Nrf2 is a key negative regulator of ferroptosis [46]. In relationship with Nrf2 function there is a target of studies, FSP1, also called “apoptosis inducing factor mitochondria associated 2”: AIFM2. As a ferroptosis inhibitor that acts independently of GPX4 to suppress ferroptosis. FSP1 acts as an oxidoreductase mainly localized on the plasma membrane and it reduces ubiquinone (coenzyme Q, or CoQ) to ubiquinol. FSP1 appears as a transcriptional target of Nrf2 and the CoQ-FSP1 axis mediates ferroptosis [47,48].

Other central cellular signaling pathways are the Hippo and transcriptional coactivator with a PSD-95/Discs large/ZO-1 (PDZ)-binding motif (TAZ) and Yes-associated protein 1 (YAP1). YAP/TAZ pathways regulate homeostasis and play a major role in regenerative processes. The Hippo pathway is regulated by the cell–cell contact and density. YAP/TAZ are sensors of the structural and mechanical features of the cell microenvironment. The YAP/TAZ activation renders cancer cells sensitive to ferroptosis. These findings establish the YAP/TAZ and Hippo pathways as novel determinants of ferroptosis [49,50]. Among the pathways modulating ferroptosis, recent findings support the notion that the induction of heat shock proteins (HSPs) helps control ferroptosis. HSPs comprise a family of molecular chaperones that are produced by cells in response to various stresses. The level of HSPB1 (a member of small HSPs) is upregulated in an HSF1 (heat shock transcription factor 1)-dependent manner. They are implicated in the regulation of iron uptake and subsequent lipid limits iron uptake in ferroptosis [51].

### 4.2. Pharmacological Approach of Ferroptosis

Molecular mechanisms associated with regulation of ferroptosis were unveiled by experiments aiming to understand the mode of action of specific ferroptotic inducers or inhibitors. Several modulators of ferroptosis have been identified through the study of ferroptotic mechanisms [52]. Ferroptosis can be induced by the canonical pathway, which is inhibition of the GPX4/GSH defense pathway, or, non-canonically, by direct stimulation of lipid peroxidation. Canonical inducers comprise class I to class III ferroptosis inducers (FIN).

#### 4.2.1. Ferroptosis Inducers

In a larger screening to find ferroptosis-inducing compounds, a series of small-molecule inducers were discovered. Induction of ferroptosis by drugs has been shown to inhibit cancer cell growth in both a Ras-dependent and -independent manner, suggesting that cancer cells display genetic heterogeneity in the control of the ferroptotic response. Drugs induce ferroptosis by inhibiting the system x_c_^-^^-^-glutathione/GPX4 axis or by regulating iron homeostasis. There are many kinds of ferroptosis-inducers, which act by the inhibition of system x_c_^-^ by inhibiting GSH, by directly inhibiting GPX4, or by altering intracellular iron levels. Generally, ferroptosis is achieved in two ways, by affecting lipid oxidation or iron metabolism. Most ferroptosis inducers can be classified on the basis of mechanism of action. Class I ferroptosis inducers indirectly reduce GPX4 function by depleting GSH. Class I ferroptosis inducers include compounds such as erastin, erastin derivatives, buthionine sulfoximine, and sorafenib [4,53]. There are two main mechanisms by which sorafenib affects ferroptosis. As with erastin, sorafenib inhibits system x_c_^−^-mediated cystine import, GSH depletion and the iron-dependent accumulation of ROS. Another mechanism is associated with the p62-Keap1-Nrf2 pathway. Nrf2 is a key regulator of the cellular antioxidant response, controlling the expression of genes that counteract oxidative and electrophilic stresses. Nrf2 is a key factor in determining the therapeutic response of ferroptosis-targeted therapies [54].

Class II inducers directly block GPX4 activity. RAS-selective lethal 3 (RSL3), a compound containing an electrophilic moiety and a chloroacetamide moiety, can react with the nucleophilic moiety of GPX4. Class III inducers, e.g., FIN56: the compound was initially CIL56 and identified as an inducer of ferroptosis for its killing activity on oncogenic RAS cells. FIN56 induces ferroptosis by two different pathways: promoting the degradation of GPX4 and reducing the abundance of CoQ10.

Recently, a few studies revealed crosstalk between distinct cell death mechanisms and antitumor immunity. The tumor microenvironment along with a variety of immune cells coexists and interplays to lead to tumorigenesis. Several in vitro and in vivo observations suggest that the activity and function of cytotoxic T cells (CD8+) and helper T cells (CD4+) are regulated by lipid peroxidation and ferroptosis. First, human naive CD4+ T cells almost lack SLC7A11, which, however, is strongly upregulated during T cell activation [55]. The activation of CD8+ T cells also has the ability to enhance the sensitivity of surrounding non-T cells to ferroptosis. Immunotherapy-activated CD8^+^ T cells enhance ferroptosis-specific lipid peroxidation in tumor cells. The increased ferroptosis contributes to the antitumor efficacy of immunotherapy [56].

#### 4.2.2. Ferroptosis Inhibitors

Ferroptosis inhibitors have also been identified. Ferroptosis can be inhibited using three main approaches: iron chelation, preventing lipid peroxidation, and scavenging lipid peroxides. Ferroptosis inhibitors exhibit various mechanisms of action, involving a reduction in ROS such as antioxidants (vitamins C, E, Trolox) and inhibitors of ROS production including ferrostatin-1, an aromatic amine that specifically binds with lipid ROS and its more potent derivatives (SRS8-72, SRS11-92, SRS12-45). The iron chelators such as desferoxamine are able to inhibit ferroptosis by reducing iron and ROS. Ebselen, a synthetic organoselenium radical scavenger compound that possesses GPX-like activity, also inhibits ferroptosis [4]. Lipid peroxidation can efficiently be blocked by the endogenous antioxidants. Increasing the cellular antioxidative activity by idebenone, a synthetic and soluble analogue of CoQ10, also reduces ferroptosis levels [57].

There are several potential points of pharmacological intervention in the ferroptosis network such as acyl-CoA synthetase long-chain family member 4.: ACSL4 drugs. ACSL4 catalyzes the esterification of PUFAs, including AA, with CoA. ACSL4 inhibitors such as rosiglitazone, troglitazone, and pioglitazone inhibit ferroptosis in a lipoxygenase-dependent manner; however, antioxidant activity also exists. Another pharmacological approach of ferroptosis prevents phospholipid peroxidation. The strategy is to introduce agents that limit the peroxidation process, for example, by preventing the propagation of lipid peroxyl radicals through the administration of lipophilic radical-trapping antioxidants, such as ferrostatin-1 as noted above or liproxstatin 1: an azaspiro compound [58]. A related approach is to administer PUFAs that are chemically resistant to peroxidation, such as by incorporation of deuterium at the bis-allylic carbon that is normally susceptible to peroxidation [59].

## 5. Factors of Modulations in Ferroptosis Process

### 5.1. Key Role of Mitochondria in Relationship with Iron Homeostasis

Ferroptotic cells are morphologically characterized by small mitochondria, collapsed mitochondrial cristae, and increased mitochondrial membrane density [4,60]. Ferroptosis is ordered by multiple proteins and cellular metabolisms, of which some are tightly associated with mitochondria. Mitochondria are cellular organelles that perform numerous bioenergetic, biosynthetic, and regulatory functions and play a central role in iron metabolism [61] and ferroptosis process [62]. As we reported previously, mitochondria represent a significant store of iron within cells, as iron is required for the functioning of respiratory chain protein complexes. It is accepted that cells use iron in mitochondria for the synthesis of heme and iron–sulfur (Fe–S) clusters. As evoked, iron metabolism is a tightly controlled process. A greater part of mitochondrial iron metabolism occurs in the mitochondrial matrix. Iron must further be transported across the inner mitochondrial membrane (IMM). This active process is dependent on membrane transporters. Mitoferrin (Mfrn1) and its homolog mitoferrin 2 (Mfrn2) are the principal importers of iron across the IMM in a variety of tissues. Mfrn1 is known to interact with Abcb10, an IMM ATP-binding cassette transporter. Mitochondria produce mitochondrial-derived peptides (MDP) that mediate the transcriptional stress response by translocating into the nucleus and interacting with deoxyribonucleic acid. This class of peptides includes humanin (HN) mitochondrial open reading frame of the 12S ribosomal ribonucleic acid type c (MOTS-c) and small humanin-like peptides. Mitochondrial-derived peptides are regulators of metabolism, exerting cytoprotective effects through antioxidative stress [63]. HN and other MDP appear to be significant in protecting cell death including apoptosis or ferroptosis in many tissues. Recent studies indicate that diverse metabolic activities through various proteins in mitochondria have active roles in inducing ferroptosis in relationship with ROS production [64].

Preventing cellular iron overload by reducing iron uptake and increasing iron storage may contribute to inhibiting ferroptosis. Additionally, it is important to remember the major role of ferritins in iron metabolism as evoked above. The ferritin family also comprises specific FtMT. FtMT is located in the mitochondrial matrix and expressed in cells with high metabolic activity in heart and brain. Mitochondrial ferritin is a H-type ferritin, suggesting that its role is to control the mitochondrial labile iron pool. FtMT can protect cells against ROS and thus against ferroptosis. Targeted induction of ferroptosis was also considered as a potential therapeutic strategy to some oxidative stress diseases, However, the precise relationship between mitochondria and ferroptosis is still matter of debate [65,66,67]. The transport of iron between cytosol, mitochondria, and lysosome, also affect the onset of ferroptosis. Lysosomes are a central position in iron homeostasis and integrate metabolic and cell death signals. A central role of the lysosome in preserving iron homeostasis and iron trafficking, is demonstrated. This function is in relationship with crosstalk including mitochondrial metabolism and ferroptosis induction [68].

### 5.2. Transforming Growth Factor-Beta (TGF-Beta) Modulations in Ferroptosis Process

Transforming growth factor-betas (TGF-betas) constitute a superfamily of multifunctional cytokines with important implications in morphogenesis, cell differentiation, and tissue remodeling. The TGF-β superfamily includes activins, bone morphogenetic proteins, the glial cell line-derived neurotrophic factor and growth/differentiation factors (GDFs). Under mitochondrial stress, cells activate specific response systems to maintain homeostasis. This mitochondrial stress response transcriptionally activates genes involved in cell survival and death. Mitochondrial stress also induces the release of distinctive secretory proteins from cells such as growth differentiation factor 15 (GDF15). This factor is a major secretory protein induced by mitochondrial dysfunction [69,70,71]. It has been hypothesized that GDF15 might play a role in erastin-induced ferroptosis by affecting the expression of ferroptosis-related genes. In a recent study, GDF15 knockdown led to decreased expression of SLC7A11, indicating that GDF15 might play important roles in ferroptosis. GDF15 knockdown promoted erastin-induced ferroptosis in a human gastric cell line [72]. GDF15 was significantly increased in neuronal ferroptosis and silencing GDF15 aggravated ferroptosis both in vitro and in vivo. In spinal cord injury mice, knockdown of GDF15 significantly exacerbated neuronal death, aggravated ferroptosis-mediated neuroinflammation, and restrained locomotor recovery [73].

### 5.3. Heme Oxygenase Modulations in Ferroptosis Process

An important regulatory process concerns the relationship between mitochondrial functions and heme oxygenase (HO) activities. Three isoforms of HO have been characterized: an inducible form (HO-1) that is upregulated, especially in the spleen and liver, in response to various types of stress, and two constitutive forms (HO-2 and HO-3). The intracellular levels of carbon monoxide (CO) can increase under stressful conditions following the induction of HO-1, an ubiquitous enzyme responsible for the catabolism of heme [17,74].

In addition to nuclear localization, evidence has accumulated for localization of HO-1 in other subcellular compartments. Specifically, a mitochondrial localization of functionally active HO-1 has been reported under stress conditions. HO-1 may be protective in modifying pro-oxidant states by preserving mitochondrial function, initiating inflammatory cell death and apoptosis. In other conditions, HO-1 releases iron as a reaction by-product, which is thereby implicated in the initiation of ferroptosis. The mechanisms by which HO-1 may impact cellular processes are multiple [74,75].

Recent studies implicate ROS-dependent signaling in the stimulation of mitochondrial biogenesis by CO through redox-dependent activation of Nrf1. A role for autophagic regulator proteins in CO-dependent cytoprotection has been reported [76]. This study uncovers a new mechanism for the protective action CO via mitochondrial ROS pathway. Moreover, beyond its widely toxicity, CO has revealed a very important biological activity as a signaling molecule with marked protective actions against apoptosis and endothelial oxidative damage [74]. Emerging evidence has revealed another “side” of HO-1, showing that HO-1 induces ferroptosis through iron accumulation. In some clinical situations, these interactions are more complex. Various studies have shown a negative action of HO-1, in which HO-1 acts as a critical mediator in ferroptosis induction and plays a causative factor for the progression of several diseases [77]. Finally, HO-1 appears as a dual regulator in iron and ROS homeostasis. HO-1 plays a dual role in ferroptosis, pro-ferroptotic and antiferroptotic effects depending on different pathological conditions and to environment conditions such as in response to the alteration in cellular redox environment.

## 6. Conclusions and Futures Directions

In conclusion, among different forms of RCD [78], ferroptosis appears to be the main cause of tissue damage driven by iron overload and lipid peroxidation. The dysfunctional ferroptotic response is implicated in a variety of pathological conditions and diseases [22]. The interrelationship between ferroptosis and cancer progression has been validated using ferroptotic agents. A series of ferroptosis inducers or inhibitors targeting redox- or iron metabolism-related proteins or signal transduction have been developed. However, to define more precise relationships between ferroptosis and health effects, it is important to define the relationship between ferroptosis and other types of cell death involving lipid peroxidation. The connection between ferroptosis and cancer progression has been validated using ferroptotic agents. Ferroptosis can be suppressed by iron chelators (e.g., deferoxamine) and lipophilic antioxidants (e.g., ferrostatin-1). The iron chelator deferoxamine is able to chelate ‘free iron’ even inside the cell. Its regular clinical use is to promote the excretion of an iron overload and to reduce ferroptosis process. According to the different targets of compounds, the ferroptosis inducer can be divided into four categories: (1) system x_c_^-^ inhibitor; (2) GPX4 inhibitor; (3) GPX4 and CoQ degradation inducer; and (4) lipid peroxide inducer.

Some small molecules (e.g., erastin and RSL3) and clinical cancer drugs (e.g., sorafenib and sulfasalazine) induced cell death in various types of cancer cells. Sorafenib is an oncogenic kinase inhibitor which has been approved as an anticancer drug in the clinical treatment. Sulfasalazine is a sulfonamide-based antimicrobial agent which is used to treat chronic inflammation of the gut; sulfasalazine inhibits breast cancer cell viability by ferroptosis [79].

Mechanisms underlying susceptibility and resistance to ferroptosis in the area of cancer have been an intense field of research in recent years. GPX4 as a cytosolic “peroxidation inhibiting protein” is a specific target of new pharmacological treatments aiming at activating or inhibiting cell death in cancer or degenerative diseases, respectively. GPX4 activity is important for the survival not only of cancer cells but also of immune cells. Various studies propose an important role of GPX4 in immunity and suggest that modulation of GPX4 expression in certain immune cells will affect antitumor immunity [80]. Bufotalin (BT), a natural small molecule, was a novel glutathione peroxidase 4 (GPX4) inhibitor, which could trigger ferroptosis in non-small-cell lung cancer cells. In vitro, BT significantly inhibited the proliferation of A549 cells and induced ferroptosis, whereas the ferroptosis inhibitor or iron chelator significantly reversed the cytotoxicity of BT on A549 cells. Furthermore, BT also inhibited tumor growth and promoted lipid peroxidation in vivo [81].

Iron is mainly taken up into cells via TfR1 which transports Tf-bound iron through receptor-mediated endocytosis. Interestingly, TfR1 was also identified and validated as a ferroptosis marker among other proposed biomarkers [82]. It is a widely used biomarker of ferroptosis in vitro or in vivo. However, it is important to consider the specificity and limitations of current biomarkers of ferroptosis [83]. Recent reports revealed that cellular energy metabolism activities such as glycolysis, pentose phosphate pathway, and tricarboxylic acid cycle are involved in the regulation of key ferroptosis markers [84]. The simultaneous detection of intracellular and extracellular markers may help diagnose and treat diseases related to ferroptotic damage. It has been proven that exosomes in body fluids can serve as biomarkers. Research on the biological role of exosomes is developing [85]. Exosomes, with an average diameter of ~100 nanometers, are a subset of extracellular vesicles. The biogenesis of exosomes involves their origin in endosomes, and interactions with other intracellular vesicles produce the ultimate content of the exosomes. Recent evidence suggests that exosomal effects involve ferroptosis. Exosomes derived from different tissues inhibit ferroptosis, which increases tumor cell chemoresistance. Moreover, exosomes have attracted attention because of their capacity as carriers of certain proteins. Exosomes can be also applied in ferroptosis-based therapy [86].

In addition, to provide an overview of ferroptosis and cancer progression, it is essential to approach the complex interplay between ferroptosis and p53 through the specific binding of p53 to the promoter of SLC7A11. A potential candidate could be H_2_O_2_ generated by natural killer cells through NADPH oxidase 2 [87]. In return, this approach is complex; p53 phenotypes being tumor type specific, applying the concept that the action of p53 in ferroptosis needs to be rationalized in a tissue-dependent manner [88]. Therefore, targeting p53/SLC7A11 signaling may be a considered therapeutic approach to reverse ferroptosis in various diseases. In recent decades, regulatory pathways and proteins have been identified which are of crucial importance for the iron and immunity interaction and their impact on microbial iron delivery. Ferroptosis has attracted considerable interest because of its potential role as a target for novel therapeutic anticancer strategies and these interactions with infections [89].

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
