# Peer review of "Lipid Peroxidation and Iron Metabolism: Two Corner Stones in the Homeostasis Control of Ferroptosis"

_ijms, 2022, doi:10.3390/ijms24010449_

Round 1

Reviewer 1 Report (Previous Reviewer 1)

I have no further suggestions

Author Response

Reviewer 2 Report (New Reviewer)

The purpose of the review is to summarize the role of lipid peroxidation and iron metabolism in the regulation of ferroptosis. A review on such an important topics should provide a well-formulated and informative summary of the current state of knowledge, which would be of interest to readers who are only partially familiar with this field. In this respect, the review is unsatisfactory.

The review presents an excessive amount of data, without ranging the data with respect of importance. The overall effect is therefore confusing, rather than enlightening.

In many cases, the review does not include appropriate references. For example, line 572 of the Discussion states: "Interestingly, TfR1 was also identified and validated as a ferroptosis marker", without citing the relevant reference. Lines 580-584 deal with exosomes, without clarifying what exosomes are (or giving a relevant reference). The legend of Figure 1 pictures the "exosome" as an intracellular organelle, rather than as a secreted vesicle.

The confusing nature of the Review can be pointed out in the section relating to iron metabolism (since this is a topic familiar with this Reviewer). Line 220 states: “Several proteins, such as transferrin, ferritins, hemosiderin, hepcidin, and ferroportin exert crucial functions in the maintenance of systemic iron homeostasis”. Hepcidin and ferroportin certainly exert a crucial function, but is it also true for hemosiderin? Very probably not, but it is certainly impossible to judge without an appropriate reference.  Line 223 states: "Iron can also be exported from the cell by FPNs. FPN molecules are expressed on the basolateral membranes of enterocytes" There is only one ferroportin protein known, not several FPNs, and ferroportin is expressed on almost all cells, not just enterocytes. Sentences like these are misleading to the reader. The cell depicted in Figure 1 is outright confusing: Since it expresses hephaestin, it should be the enterocyte, but enterocytes do not acquire iron by transferrin receptors. Again, a review should be enlightening, not confusing. The problem is probably not restricted to iron metabolism: Line 98 states "ROS are formed when molecular oxygen is reduced to water". In this Reviewer understanding, reduction of oxygen to water produces water, while ROS are formed by one-electron reductions (?) Line 540 states: "The connection between ferroptosis and cancer progression has been validated using ferroptotic agents. Ferroptosis can be suppressed by iron chelators and lipophilic antioxidants and inhibition of ferroptosis could diminish related clinical symptoms". This appears to be a wrong statement, since the great expectation for ferroptosis manipulation is to induce it in cancer cells.  

The use of English language requires further poofreading. Phrases like "As we will develop later in this review" (line 248), "Hepcidin is an antimicrobial peptide with iron regulatory properties, was discovered and it plays... (line 255), or "Although this kind of RCD has recently attracted great interest in basic and clinical research." (line 536) are confusing.

The title of the Review is confusing as well, as it implies that both lipid peroxidation and iron metabolism control ferroptosis and ferritinophagy. The role of lipid peroxidation in the control of ferroptosis is evident, but no evidence is given for its control of ferritinophagy.

Overall, the general impression is that the Review does not give the reader a concise summary of the state of art knowledge on ferroptosis, and that it does not improve on the many reviews on ferroptosis already written. As such, it can probably not be very useful to potential readers.

Round 2

Reviewer 2 Report (New Reviewer)

The authors have partially responded to the questions raised. However, some issues remain, and need further clarification.

In line 98, the authors have not corrected the misleading sentence "ROS are formed when molecular oxygen is reduced to water." Reduction of oxygen to water is a two electron process which does NOT produce ROS. Of course, in mitochondria, partial production of ROS takes place, but this has to be explained.

Figure 1 is confusing. The annotation of exosome as a "secreted vesicle" did not clarify matters, nor did the deletion of hephestin protein. Do the authors suggest that iron from LIP enters the exosome? And do they suggest that intracellular ferritin is localized in exosomes? The sentence on line 268 seems to suggest this. Does ferritinophagy occur in exosomes? The Figure seems to indicate this. And, after export from the cell, how is ferrous iron oxidised to ferric iron? The Figure suggests that this is a heme oxygenase-mediated process. In conclusion, the Figure should be redrawn.

Paragraph 5.2. should address the role of TGF beta in modulating ferroptosis. The connection to ferroptosis is unclear, ferroptosis is not once mentioned in the paragraph. 

Round 3

Reviewer 2 Report (New Reviewer)

The authors partially responded to the comments, and corrected the two issues related to free radical generation and the role of GDF15.  However, Figure 1 still needs to be redrawn. The authors did NOT answer the raised questions related to the Figure, namely: 

1). Does the Figure mean that that iron from LIP enters the exosome?

2). Does the Figure mean that intracellular ferritin is localized in exosomes?

3). Does ferritinophagy occur in exosomes?

The Figure seems to indicate all these situations. Unless the authors have definite proof for these statements, it needs to be completely redesigned. 

Author Response

This manuscript is a resubmission of an earlier submission. The following is a list of the peer review reports and author responses from that submission.

Round 1

Reviewer 1 Report

In this review the authors deal with an interesting theme concerning a type of cell death: ferroptosis, that is very interesting and current as well as important for its consequences on clinic and in therapy.

The review highlights important aspects of the topic, however there are some inaccuracies that need to be corrected. Also in the text there are many repetitions that could be eliminated for  better reading.

Major points:

Lane 62:

The authors stated “This novel form of regulated cell death (RCD), distinct from apoptosis and necrosis was coined in 2012” and in lane 75 “Ferroptosis is considered as a type of ROS-dependent regulated necrosis” and this concept was repeated also in lane 210. This statement is somewhat confusing, ferroptosis has been established to be a type of necrosis. The concept described in lane 62 needs to be better explained.

Lane 246-248.

This sentence does not mention the reference that reported the data, however I think it refers to the work of Fang X. et al 2020. As it is written, it implies that it is ferritin that has a direct action on the down regulation of SLC7A11, while probably the p53 signaling pathway is involved. In my knowledge there no data on the direct involvement of ferritin in the regulation of the expression of SLC7A11. The sentence must be rephased as well as the one in lane 343.

Lane 450

IMM is not a process, please correct the sentence.

Lane 566

The TF receptor transport essentially Transferrin-bound iron and not ferritin, please correct.

Fig. 1 the figure shows the Fe3 + transported by the TfR / Tf complex in the cytosol, this is  incorrect, the iron is released from the Tf in the endosome, where it is reduced to Fe2+ and not released directly in the cytosol. Please modify the picture.

Minor points:

The acronyms of iron proteins have been defined by a panel of international experts, for clarity it would be better to identify ferritins as FtH, FtL, FtMt, the transferrin receptor as TfR and transferrin as Tf.

Text should be checked for punctuation and repetition.

Reviewer 2 Report

Rochette et al.,

This reviewer has read with great interest the review entitled Lipid peroxidation and iron metabolism: two corner stones in 2 the homeostasis control of ferritinophagy and ferroptosis. However, overall the review seems to have been written hastily and does not sufficiently enhance the topics covered which are of great interest today. There is also a need for in-depth editing and even adding more graphics to make the style more interesting.

Sometimes there are incomplete descriptions. Some examples: 1. three types of death are mentioned but not all are described (rows 52-53); RNS species are listed but ROS are not (rows 88…), some ROS species are detailed later (row 116); and the link between lipid peroxidation and immunogenic death, although poorly understood, is poorly expressed (rows 143..). Paragraph 2.1 should be restructured and reframed.

Paragraph 4: 4.1 refers to cellular pathways. What does text in row 308 do?

Paragraph 5. Potential biological functions of ferroptosis: please change this title. This paragraph is poorly focused and information is broken down into small pieces that are not pleasant to read and understand.

Text related to conclusions and future directions is poor and should be implemented.

At this stage the review clearly lacks major outcomes that would justify publication.

Reviewer 3 Report

As a reviewer, I am always trying to write a constructive review, to help the authors to improve their manuscript. I have approached the present manuscript with the same attitude, and I attempted to point out factual errors, or unclear phrases, as well as list out editing mistakes (syntax, punctuation, typos). However, the longer I was reading, the less doable this became to list points that should be corrected or clarified. The review seems to be written not in a careful manner, with many errors, both scientific (this I can certainly say for the part that relates to iron metabolism) and language-related. The order of information is quite chaotic, some phrases or players appear without a clear and well-structured introduction, while, paradoxically, some pieces of information are repetitive. Critical references are lacking next to the cited original findings, and some key papers in the field are not cited at all (eg, Friedmann Angeli 2014 NSB; while there are quite a few auto-citations). In its current state, I find the manuscript not good enough to be easily revised, it would need to be re-written.

Round 2

Reviewer 1 Report

The authors satisfied my requests

Reviewer 2 Report

The manuscript does not reach the required quality standards.

Reviewer 3 Report

The authors have not address my concerns. The manuscript still contains many language-related and editing errors. It still requires in-depth English editing to be of sufficient quality for publication. Regarding scientific errors - the part related to iron metabolism has incorrect information and shows that the authors are not familiar with the field. It is not the role of the reviewer to edit and correct the content of the review manuscript. The authors shall consider inviting another co-author to this review, if offered another major revision, for a thorough revising of this part.